# Association between Body Roundness Index and Depression Among Middle-aged and Older Adults in Chinese Communities: An Empirical Analysis Based on CHARLS Data

**Wenfei Yang, Liping Chen, Liling Tong, Wenchang He, Hua Lin** ⓘ*

Department of Emergency, Haikou Affiliated Hospital of Central South University Xiangya School of Medicine, Haikou, Hainan, China

* 1264615433@qq.com

## Abstract

### Background

The relationship between depression and obesity has been confirmed by multiple studies. Compared to conventional measurement indicators such as body mass index or waist circumference, the body roundness index (BRI) demonstrates higher accuracy in assessing body fat content, especially visceral adiposity. Nevertheless, despite the advantages of BRI in measuring fat, the specific link between BRI and depression remains unclear. This study aims to clarify the potential correlation using data from the China Health and Retirement Longitudinal Study (CHARLS).

### Methods

This study used CHARLS data from 2015 and 2020. We screened and included 7,258 middle-aged and older adults without depressive symptoms at baseline. We explored the connection between BRI and depression risk through logistic regression analyses, restricted cubic spline analyses, subgroup analyses, and interaction tests

### Results

After adjusting for covariates, a positive correlation was observed between BRI and depression risk. Specifically, a one-unit increase in BRI led to a 14% increase in depression risk (OR = 1.14, 95% CI: 1.09-1.20, $P < 0.001$).

### Conclusion

BRI is linked to a higher risk of depression in middle-aged and older adults in China and can be used as a simple indicator to predict depression.

**Data availability statement:** All relevant data are within the paper and its Supporting Information files.

**Funding:** This study was supported by the Hainan Province Department of Science and Technology Development Project (Project Number: 822RC864). The funders had no role in study design, data collection and analysis,

decision to publish, or preparation of the manuscript.

**Competing interests:** The authors have declared that no competing interests exist.

## 1. Introduction

Depression and obesity are global health challenges that impact public health. The core symptoms of depression are significant and persistent low mood, accompanied by impaired cognitive function, physical symptoms, and reduced social function. In more serious cases, patients may even have suicidal thoughts [1]. Depression affects patients' quality of life, social functioning, and creates a large economic burden for families and society [2]. The World Health Organization reports that about 350 million people global suffer from depression, with an increasing incidence rate yearly [3]. In China, about 33.1% of adults aged 60 and above show depression symptoms, with a prevalence of major depressive disorder reaching 5.3% [4]. Elderly people are more likely to suffer from depression due to declining physiological functions, social role changes, and increased chronic disease rates [5]. The causes of depression involve genetics, neurobiology, and social influences [6,7]. Additionally, depression exhibits high heterogeneity and uncertainty in symptoms, progression, prognosis, and treatment response, making treatment and management challenging [8]. Therefore, strengthening early identification and prevention of depression is crucial for effectively curbing the disease progression.

Evidence shows that obesity is an important risk factor for depression. Obese individuals have a 1.29 to 1.69 times higher risk of depression than those with normal weight [9]. Traditionally, Body Mass Index (BMI) and Waist Circumference (WC) are commonly used to assess obesity, but both WC and BMI have their limitations in evaluating obesity. BMI cannot distinguish fat distribution from muscle mass, and WC as an indicator of obesity assessment also has its inherent limitations [10]. In addition, there is a complex nonlinear association between obesity and mortality, called the "obesity paradox" [11], which challenges the application of traditional obesity indicators in health risk assessment. Studies on the association between obesity and depression show complexity, with some suggesting obesity increases depression risk [12] and others suggesting a protective effect [13]. This inconsistency highlights the need for a more detailed evaluation of traditional indicators like BMI and calls for the exploration of more accurate methods for measuring obesity phenotypes.

The body roundness index (BRI) is a new measure of obesity that combines WC and height, providing a more accurate reflection of abdominal fat accumulation and individual body characteristics [14]. Compared to BMI, BRI can more accurately assess the severity of obesity [15]. Studies have demonstrated that BRI can effectively forecast the risk of chronic conditions [16,17]. A study based on the National Health and Nutrition Examination Survey (NHANES) revealed a positive correlation between BRI and depression in adults [18], but the applicability of this conclusion in middle-aged and older adults still requires further validation. Given the accelerated aging of China's population and the rising prevalence of depression among middle-aged and older adults, it is important to explore the connection between BRI and depressive symptoms in this demographic. Therefore, this research seeks to explore the relationship between BRI and depression using data from the China Health and Retirement Longitudinal Study (CHARLS), providing scientific evidence for the prevention and intervention of obesity and depression.

## 2. Materials and methods

### 2.1 Data source

CHARLS is a longitudinal survey project that collected microdata on individuals aged 45 and above and their families in China, allowing for in-depth analysis of China's aging trend, promoting interdisciplinary research on aging issues, and offering a solid basis for policy-making. The first baseline survey was conducted in 2011, covering 28 provinces across the

country with over 17,000 participants. The survey utilized a multi-stage probability proportional to size (PPS) sampling technique to guarantee the representativeness. The personal survey encompasses multiple aspects, including demographic information, family structure, health status, utilization of medical services, and community environment. Typically, CHARLS conducts follow-up interviews every two to three years, and five rounds of data collection have been completed to date. CHARLS data is publicly available through its official website, and users can download and use it (https://charls.pku.edu.cn/). The project was approved by the Biomedical Ethics Committee of Peking University. The ethical review approval numbers for CHARLS, including anthropometric measurements and biomarker collection, were IRB00001052-11015 and IRB00001052-11014, respectively. The study strictly adheres to the STROBE Statement (Strengthening the Reporting of Observational Studies in Epidemiology) guidelines for reporting and provides a comprehensive disclosure of the key details of the study design, participant selection, data collection, and analysis methods [19].

This study utilized data from the third wave (2015) and fifth wave (2020) of the CHARLS database. The baseline survey in 2015 included 21,038 participants. Based on the following criteria, we excluded 11,371 individuals: (1) missing or abnormal data; (2) age < 45; (3) presence of cognitive impairment (such as dementia, intellectual disability, or mental/psychological problems); (4) absence of covariates; (5) baseline depression symptoms, missing or abnormal height or WC data. After screening, a total of 9,667 participants met the inclusion criteria. In the subsequent longitudinal analysis, we excluded individuals who were lost to follow-up or had died in the fifth wave survey, as well as participants with missing depression score data, resulting in a final sample of 7,258 eligible individuals. The detailed screening process is shown in Fig 1. For data that were significantly abnormal compared to measurements from other years (e.g., height of 95 cm, WC of 13 cm), we treated them as missing values.

## 2.2 Measurement of depression

In this study, the Center for Epidemiologic Studies Depression Scale (CESD-10) was employed as a tool to assess participants' depressive symptoms. CESD-10 is widely employed among the elderly to measure their depression status. Consisting of 10 items, each with 4 options and a scoring range of 0 to 3 points, the fifth and eighth items are scored in reverse. The total score spans from 0 to 30 points, with higher scores demonstrating a more pronounced tendency towards major depressive disorder. This study used a CESD-10 score of ≥ 10 as the diagnostic criterion for depression [20]. Prior research has confirmed that CESD-10 is reliable and effective [21]. In comparison to other depression measurement scales, CESD-10 is more practical [22]. Consequently, CESD-10 is an essential part of CHARLS research and is widely used in Chinese populations [23].

## 2.3 Measurement of BRI

BRI evaluates body shape by measuring height and WC. The height and WC data for this study were obtained from participants' baseline physical examinations and measured by professionals. To ensure accuracy, participants needed to remove their shoes and stand while their height was measured. WC was measured by placing a tape measure horizontally around the waist at the level of the navel. The formula for calculating BRI is as follows:

$$BRI = 364.2 - 365.5 \times \sqrt{1 - \frac{(WC\ cm\ /2\pi)^2}{(0.5 \times H\ cm)^2}}$$ , where WC represents waist circumference and H

represents height [14].

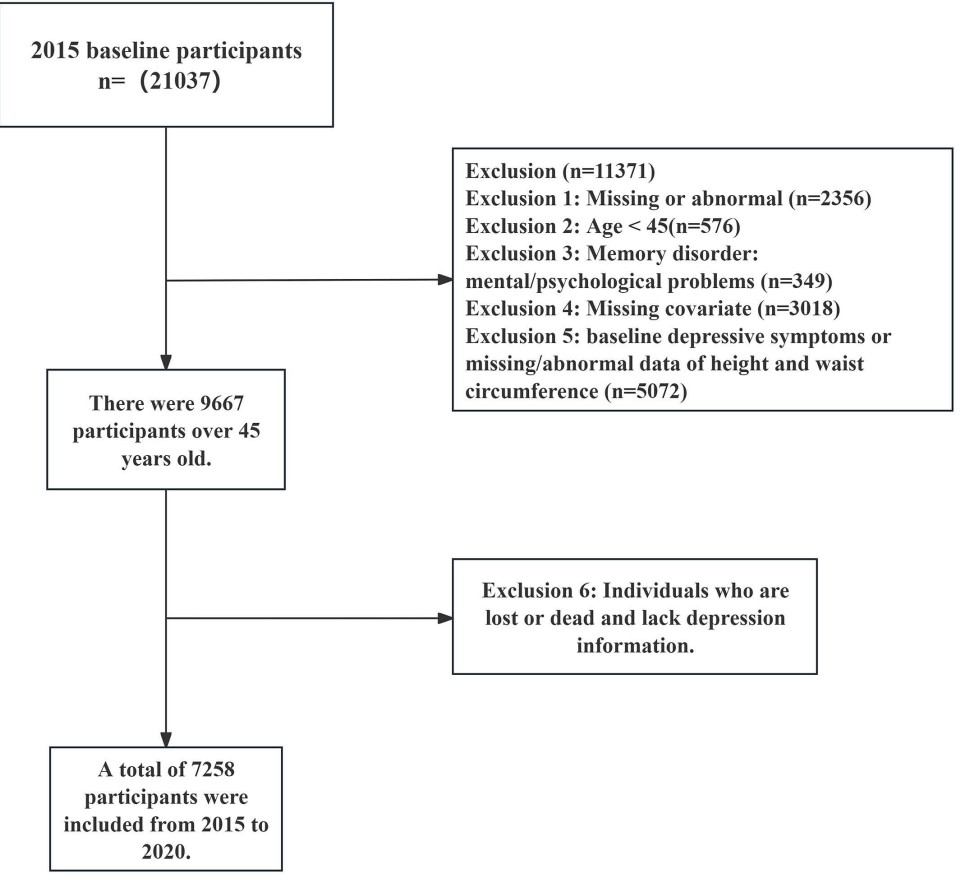

**Fig 1. Flow chart of research sample inclusion.**

## 2.4 Measurement of Covariates

Covariates included sociodemographic factors, lifestyle behaviors, and health status. Specifically, sociodemographic factors included age, gender, residence (urban, rural), educational level (primary school and below, middle school, high school, college degree and above), marital status (married, Unmarried divorce and other), life satisfaction (extremely satisfied, very satisfied, satisfied, not very satisfied), and social participation (yes, no). Lifestyle behaviors included smoking status, alcohol consumption, sleep duration ($\leq$ 6 hours, 6-8 hours, $\geq$ 8 hours). Health status was evaluated based on the presence of pain and chronic diseases such as hypertension, diabetes, stroke, cancer, cardiopathy, and kidney disease.

## 2.5 Statistical Analysis

Data cleaning and analysis were performed by R software (version 4.4.1). Initially, based on the baseline data from 2015 (wave 3), the BRI was divided into quartiles. The differences in the basic features of each group in the first wave of data were evaluated using analysis of variance and chi-square tests. Categorical variables are expressed as frequencies and percentages, while continuous variables are presented as means and standard deviations. To examine the relationship between BRI and depression risk, logistic regression analysis was conducted, and three progressively adjusted models were developed. Model 1: Included only BRI as the sole explanatory variable in univariate logistic regression. Model 2: Adjusted for sociodemographic

factors like gender, age, educational level, marital status, and residence, building on Model 1. Model 3: Further incorporated health-related factors, such as life satisfaction, sleep duration, social activity, pain, chronic diseases (hypertension, diabetes, etc.), and smoking and drinking habits. To investigate the possible nonlinear association between BRI and depression risk, a restricted cubic spline (RCS) analysis was performed, and reference values were set. Additionally, a two-stage linear regression model was utilized to identify the threshold effect of BRI on depression risk. The optimal model was chosen based on the Akaike's information criterion (AIC). Subgroup analyses were conducted, stratified by age, gender, presence of chronic diseases, and smoking and drinking behaviors. Interaction effects were assessed using P-values to evaluate consistency across subgroups, and the log-likelihood ratio test was used to determine the significance of these interactions. The significance level for all statistical tests is set at $P <$ 0.05.

## 3. Results

### 3.1 Baseline Characteristics

At wave 1, a total of 7,258 individuals were classified as non-depressed patients. According to the BRI score, these participants were divided into four groups: Q1 (BRI < 3.396), Q2 (3.396 ≤ BRI < 4.272), Q3 (4.272 ≤ BRI < 5.194), and Q4 (BRI ≥ 5.194). These groups showed significant differences ($P <$ 0.05) in multiple aspects, including sociodemographic factors (age, gender, residence, education level, marital status, life satisfaction, and social participation), lifestyle behaviors (smoking status, alcohol consumption, and sleep duration), and health status (presence or absence of pain and the diagnosis of chronic diseases such as hypertension, diabetes, and heart disease). Compared to the Q1 group, the Q4 group exhibited a higher proportion of females and urban residents. Additionally, the Q4 group had a higher education level and more participation in social activities. In terms of lifestyle factors, the Q4 group had a lower smoking rate but a higher alcohol consumption rate. In terms of health status, the prevalence of diabetes, heart disease, hypertension, and pain symptoms was notably higher in the Q4 group in comparison to the Q1 group (Table 1). These results suggest that BRI may be closely related to lifestyle choices, socioeconomic status, and overall health status in middle-aged and older adults.

### 3.2 Association between BRI and Depression Risk

The results of logistic regression analysis indicated a notable positive association between BRI and depressive symptoms (OR = 1.14, 95% CI: 1.09-1.20, $P <$ 0.001), which persisted after adjusting for various covariates in models 1, 2, and 3. Additionally, stratified analysis based on BRI quartiles indicated that individuals in the highest quartile (Q4: ≥ 5.194) faced a greatly higher risk of depression in comparison to those in the lowest quartile (Q1: < 3.396) (OR = 1.39, 95% CI: 1.16-1.65, $P <$ 0.001). Notably, in model 3, each unit increase in BRI was linked to a 14% elevation in depression risk (Table 2).

### 3.3 RCS Analysis of the Association between BRI and Depression Risk

To investigate the nonlinear relationship between BRI and depression risk, we conducted the RCS analysis and selected the optimal model between 3 and 7 knots based on the AIC. Ultimately, the 4-knot model with the lowest AIC value was chosen for subsequent analysis. The RCS analysis results showed a significant overall trend (overall $P <$ 0.0001) between BRI and the risk of depression, and a significant nonlinear association was observed (nonlinear $P$ = 0.004). The specific results are shown in Fig 2 and Table 3. Further analysis of the threshold effect indicated that the inflection point of BRI was 4.29. Segmented logistic regression

**Table 1. Comparison of basic characteristics and quartile BRI of middle-aged and elderly people in China.**

| Variables | Level | Overall | Q1 | Q2 | Q3 | Q4 | *P*-value |
|---|---|---|---|---|---|---|---|
| | | 7258 | 1815 | 1814 | 1814 | 1815 | |
| Age [IQR] | | 58.0 [51.0, 65.0] | 58.0 [51.0, 65.0] | 57.0 [51.0, 64.0] | 58.0 [51.0, 64.0] | 59.0 [52.0, 65.0] | <0.001 |
| Gender, n (%) | | | | | | | <0.001 |
| | Female | 3415 (47.1) | 539 (29.7) | 784 (43.2) | 907 (50.0) | 1185 (65.3) | |
| | Male | 3843 (52.9) | 1276 (70.3) | 1030 (56.8) | 907 (50.0) | 630 (34.7) | |
| Marital status, n (%) | | | | | | | 0.058 |
| | Unmarried divorce and others | 583 (8.0) | 156 (8.6) | 119 (6.6) | 149 (8.2) | 159 (8.8) | |
| | Married | 6675 (92.0) | 1659 (91.4) | 1695 (93.4) | 1665 (91.8) | 1656 (91.2) | |
| Educational leve, n (%) | | | | | | | 0.005 |
| | Primary school and below | 2434 (33.5) | 635 (35.0) | 589 (32.5) | 594 (32.7) | 616 (33.9) | |
| | Middle school | 2200 (30.3) | 566 (31.2) | 585 (32.2) | 561 (30.9) | 488 (26.9) | |
| | High school | 1659 (22.9) | 369 (20.3) | 415 (22.9) | 417 (23.0) | 458 (25.2) | |
| | College degree and above | 965 (13.3) | 245 (13.5) | 225 (12.4) | 242 (13.3) | 253 (13.9) | |
| Residence, n (%) | | | | | | | <0.001 |
| | Urban | 2958 (40.8) | 589 (32.5) | 699 (38.5) | 833 (45.9) | 837 (46.1) | |
| | Rural | 4300 (59.2) | 1226 (67.5) | 1115 (61.5) | 981 (54.1) | 978 (53.9) | |
| Drinking, n (%) | | | | | | | <0.001 |
| | Yes | 4368 (60.2) | 954 (52.6) | 1068 (58.9) | 1071 (59.0) | 1275 (70.2) | |
| | No | 2890 (39.8) | 861 (47.4) | 746 (41.1) | 743 (41.0) | 540 (29.8) | |
| Smoking (%) | | | | | | | <0.001 |
| | Yes | 5062 (69.7) | 974 (53.7) | 1246 (68.7) | 1339 (73.8) | 1503 (82.8) | |
| | No | 2196 (30.3) | 841 (46.3) | 568 (31.3) | 475 (26.2) | 312 (17.2) | |
| Sleeping hours, n (%) | | | | | | | 0.925 |
| | ≤6 | 2912 (40.1) | 724 (39.9) | 742 (40.9) | 730 (40.2) | 716 (39.4) | |
| | 6～8 | 3607 (49.7) | 896 (49.4) | 897 (49.4) | 904 (49.8) | 910 (50.1) | |
| | ≥8 | 739 (10.2) | 195 (10.7) | 175 (9.6) | 180 (9.9) | 189 (10.4) | |
| Social participation, n (%) | | | | | | | <0.001 |
| | No | 3003 (41.4) | 836 (46.1) | 751 (41.4) | 700 (38.6) | 716 (39.4) | |
| | Yes | 4255 (58.6) | 979 (53.9) | 1063 (58.6) | 1114 (61.4) | 1099 (60.6) | |
| Pain, n (%) | | | | | | | 0.022 |
| | No | 6062 (83.5) | 1523 (83.9) | 1539 (84.8) | 1525 (84.1) | 1475 (81.3) | |
| | Yes | 1196 (16.5) | 292 (16.1) | 275 (15.2) | 289 (15.9) | 340 (18.7) | |
| Life satisfaction, n (%) | | | | | | | 0.002 |
| | Extremely satisfied | 241 (3.3) | 78 (4.3) | 54 (3.0) | 65 (3.6) | 44 (2.4) | |
| | Very satisfied | 3434 (47.3) | 892 (49.1) | 848 (46.7) | 867 (47.8) | 827 (45.6) | |
| | Satisfied | 3012 (41.5) | 686 (37.8) | 779 (42.9) | 742 (40.9) | 805 (44.4) | |
| | Not very satisfied | 571 (7.9) | 159 (8.8) | 133 (7.3) | 140 (7.7) | 139 (7.7) | |
| Hypertension, n (%) | | | | | | | <0.001 |
| | No | 5450 (75.1) | 1573 (86.7) | 1460 (80.5) | 1314 (72.4) | 1103 (60.8) | |
| | Yes | 1808 (24.9) | 242 (13.3) | 354 (19.5) | 500 (27.6) | 712 (39.2) | |
| Diabetes, n (%) | | | | | | | <0.001 |
| | No | 6751 (93.0) | 1767 (97.4) | 1692 (93.3) | 1693 (93.3) | 1599 (88.1) | |
| | Yes | 507 (7.0) | 48 (2.6) | 122 (6.7) | 121 (6.7) | 216 (11.9) | |
| Cancer, n (%) | | | | | | | 0.124 |
| | No | 7179 (98.9) | 1791 (98.7) | 1794 (98.9) | 1803 (99.4) | 1791 (98.7) | |
| | Yes | 79 (1.1) | 24 (1.3) | 20 (1.1) | 11 (0.6) | 24 (1.3) | |
| Cardiopathy, n (%) | | | | | | | <0.001 |

*(Continued)*

**Table 1.** (Continued)

| Variables | Level | Overall | Q1 | Q2 | Q3 | Q4 | P-value |
|---|---|---|---|---|---|---|---|
| | No | 6390 (88.0) | 1681 (92.6) | 1617 (89.1) | 1596 (88.0) | 1496 (82.4) | |
| | Yes | 868 (12.0) | 134 (7.4) | 197 (10.9) | 218 (12.0) | 319 (17.6) | |
| Stroke (%) | | | | | | | 0.095 |
| | No | 7122 (98.1) | 1792 (98.7) | 1781 (98.2) | 1771 (97.6) | 1778 (98.0) | |
| | Yes | 136 (1.9) | 23 (1.3) | 33 (1.8) | 43 (2.4) | 37 (2.0) | |

Abbreviations: BRI, body roundness index; SD, standard deviation; OR, Odds Ratio; 95%CI, 95% Confidence Interval

analysis revealed no dramatic association when BRI was below 4.29 (OR = 0.99, 95% CI: 0.91-1.07, $P$ = 0.864). However, when BRI exceeded 4.29, the risk of depression notably increased by 20% (OR = 1.20, 95% CI: 1.13-1.27, $P$ < 0.0001).

P for overall < 0.05 indicates the correlation between indicators and outcomes.

P for non-linear < 0.05 indicates that there is a nonlinear relationship between indicators and outcomes.

BRI is non-linearly associated with depressive risk. When BRI is less than 4.29, it has no significant impact on depressive risk; whereas when BRI exceeds 4.29, depressive risk significantly increases.

### 3.4 Subgroup Analysis

Subgroup analyses were conducted to explore the connection between BRI and depression risk. Participants were categorized based on age, gender, residence, smoking status, and the presence of chronic diseases. The results indicated that, with the exception of individuals aged 70 and above and those diagnosed with cancer, the association between all other subgroups remained significant ($P$ < 0.05). This suggests strong consistency in the relationship across the various groups (Fig 3).

## 4. Discussion

The results of this study demonstrate a significant association between BRI and depression risk. Even after adjusting for variables like gender, age, lifestyle, and chronic diseases, this association remains significant. Specifically, higher BRI is closely linked to increased depression risk in middle-aged and older adults, with each unit increase in BRI associated with a 14% higher risk of depression. Subgroup analysis and interaction tests confirmed this association's universal applicability across different demographics and health conditions, making BRI a powerful predictive tool for assessing depression risk in this population.

Obesity is a complex chronic disease often associated with health issues like cardiovascular disease (CVD) and mental disorders. Research has confirmed a significant correlation between abdominal obesity and depression in adults [24]. WC, a measure of abdominal obesity, is also associated with the risk of depression. However, Wade-Bohleber et al.[25] pointed out that assessing depression risk based on abdominal obesity is more accurate than using general body fat content. Nevertheless, existing studies generally only use WC to measure abdominal obesity without fully considering height factors, which may lead to misleading conclusions in populations with significant height differences [26]. Therefore, this study aims to explore in depth the association between BRI and the risk of depression. As an indicator that comprehensively considers both height and WC, BRI can more accurately reflect an individual's body fat ratio and obesity status. For example, a study involving 18,654 adults aged 20 and above demonstrated a notable positive link between BRI and the incidence of

**Table 2.  Correlation analysis between BRI and the risk of depression.**

|  | Model 1[a] | | | Model 2[b] | | | Model 3[c] | | |
|---|---|---|---|---|---|---|---|---|---|
|  | OR | 95%CI | P-value | OR | 95%CI | P-value | OR | 95%CI | P-value |
| BRI | 1.12 | 1.08, 1.16 | <0.001 | 1.14 | 1.10, 1.20 | <0.001 | 1.14 | 1.09, 1.20 | <0.001 |
| BRI4 |  |  |  |  |  |  |  |  |  |
| Q1 | — | — |  | — | — |  | — | — |  |
| Q2 | 0.96 | 0.83, 1.11 | 0.614 | 0.92 | 0.78, 1.09 | 0.354 | 0.92 | 0.77, 1.09 | 0.313 |
| Q3 | 1.12 | 0.97, 1.30 | 0.109 | 1.1 | 0.93, 1.30 | 0.266 | 1.1 | 0.92, 1.30 | 0.298 |
| Q4 | 1.38 | 1.20, 1.59 | <0.001 | 1.41 | 1.19, 1.67 | <0.001 | 1.39 | 1.16, 1.65 | <0.001 |

[a]Model 1: was adjusted for no covariates;

[b]Model 2: Adjust for the variables in model 1 plus gender, age, education level, marital status and place of residence.

[c]Model 2: Adjust for the variables in model 2 plus life satisfaction, sleep time, social activities, pain, hypertension, diabetes, stroke, cancer, coronary heart disease, smoking and drinking.

Abbreviations: BRI, body roundness index; SD, standard deviation; OR, Odds Ratio; 95%CI, 95% Confidence Interval

depression [18]. In a model that comprehensively adjusted for age- and gender-related factors, for every unit increase in BRI, the prevalence of depression significantly rose by 8%. Another community survey based on NHANES showed that in the female group, the incidence rate of depression increased by 14% for each unit increase in BRI [27]. However, current research on the relationship between BRI and depression primarily focuses on Western populations, with relatively little attention paid to the middle-aged and older adults in China. As one of the countries facing the most severe aging issues globally, the physical and mental health concerns of China's elderly population are in urgent need of attention. Our results indicate a dramatic positive correlation between BRI and depression risk among middle-aged and older adults in China. This finding is consistent with previous research conducted in American adults [18], further confirming the broad association between BRI as an obesity indicator and the risk of depression. By extending this association to the middle-aged and older adults in China, we provides a new perspective on understanding the relationship between BRI and depression in different ethnic and cultural backgrounds. In summary, this study not only fills the gap in research on the association between BRI and depression in middle-aged and older adults, but also provides valuable epidemiological evidence for the prevention and intervention of depression in China and potentially globally in response to the aging population.

Obesity can potentially escalate the risk of depression among middle-aged and older adults through several mechanisms. Firstly, with age, metabolic functions and antioxidant capacities typically diminish. The fat accumulation resulting from obesity further propels the production of reactive oxygen species (ROS). When ROS levels are excessive and the body's antioxidant capacity is insufficient, inflammatory signaling pathways can be activated, leading to damage of biomolecules and inducing cell apoptosis [28,29]. Moreover, the inflammatory responses triggered by obesity may intensify common neurodegenerative diseases in the elderly, such as Alzheimer's and Parkinson's diseases, thereby augmenting the risk of depression [30]. Secondly, both depression and obesity share similar risk factors, one of which is an imbalance in gut microbiota. The gut microbiota can communicate with the brain, forming the gut-microbiota-brain axis. Obesity and high-fat diets can alter the gut microbiota, which in turn affects human emotions and behavior through this axis [31]. Thirdly, obesity may trigger overactivity in the hypothalamic-pituitary-adrenal (HPA) axis, causing an excessive release of cortisol. Prolonged high levels of glucocorticoids can impair the function of the HPA axis, further contributing to the development and progression of depression [32]. Fourthly, prolonged excessive energy intake can disrupt synaptic plasticity, leading to alterations in neuronal

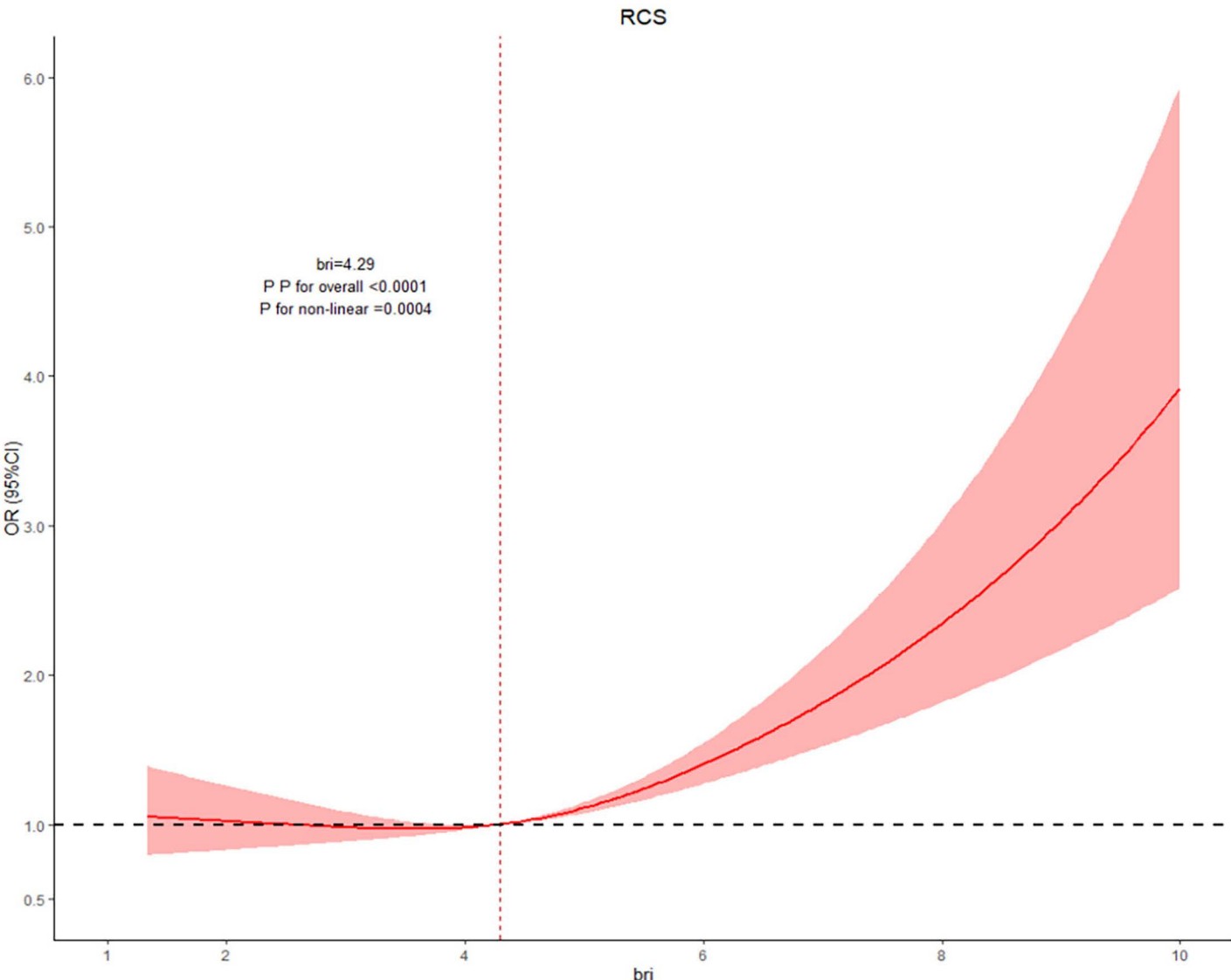

**Fig 2. RCS diagram of the association between BRI and the risk of depression.**

**Table 3. Threshold Effect Analysis of BRI on Depression Based on Two-stage Linear Regression Model.**

| Depression | Adjust OR (95% CI) | *P*-value |
|---|---|---|
| BRI | 1.14(1.09, 1.20) | <0.001 |
| Fitting by standard linear model | | |
| Fitting by two-piecewise linear model | | |
| Inflection point | 4.29 | |
| <4.29 | 0.99(0.91, 1.07) | 0.864 |
| >4.29 | 1.20(1.13, 1.27) | <0.0001 |
| Log-likelihood ratio | <0.001 | |

Abbreviations: BRI, body roundness index; SD, standard deviation; OR, Odds Ratio; 95%CI, 95% Confidence Interval

| Variable | Count | Percent(%) | P.value | P.for.interaction | | OR (95% CI) |
|---|---|---|---|---|---|---|
| gender | | | | 0.392 | | |
| female | 3415 | 47.1 | <0.001 | | | 0.88 (0.83 to 0.92) |
| male | 3843 | 52.9 | <0.001 | | | 0.90 (0.86 to 0.96) |
| age | | | | 0.238 | | |
| 45-60 | 4087 | 58.2 | <0.001 | | | 0.87 (0.83 to 0.91) |
| 61-70 | 2205 | 31.4 | 0.002 | | | 0.90 (0.85 to 0.96) |
| ≥71 | 734 | 10.4 | 0.397 | | | 0.96 (0.86 to 1.06) |
| Residence | | | | 0.242 | | |
| Rural | 2958 | 40.8 | 0.003 | | | 0.92 (0.87 to 0.97) |
| Urban | 4300 | 59.2 | <0.001 | | | 0.88 (0.84 to 0.92) |
| Hypertension | | | | 0.143 | | |
| No | 5450 | 75.1 | <0.001 | | | 0.91 (0.87 to 0.95) |
| Yes | 1808 | 24.9 | <0.001 | | | 0.86 (0.80 to 0.92) |
| diabetes | | | | 0.759 | | |
| No | 6751 | 93 | <0.001 | | | 0.90 (0.86 to 0.93) |
| Yes | 507 | 7 | 0.043 | | | 0.88 (0.77 to 1.00) |
| Cance | | | | 0.711 | | |
| No | 7179 | 98.9 | <0.001 | | | 0.89 (0.86 to 0.93) |
| Yes | 79 | 1.1 | 0.682 | | | 0.94 (0.71 to 1.25) |
| Cardiopathy | | | | 0.208 | | |
| No | 6390 | 88 | <0.001 | | | 0.90 (0.87 to 0.94) |
| Yes | 868 | 12 | 0.001 | | | 0.84 (0.76 to 0.93) |
| Stroke | | | | 0.265 | | |
| No | 7122 | 98.1 | <0.001 | | | 0.90 (0.86 to 0.93) |
| Yes | 136 | 1.9 | 0.064 | | | 0.76 (0.56 to 1.02) |
| drinking | | | | 0.226 | | |
| No | 4368 | 60.2 | <0.001 | | | 0.88 (0.85 to 0.92) |
| Yes | 2890 | 39.8 | 0.019 | | | 0.93 (0.87 to 0.99) |
| Smoking | | | | 0.874 | | |
| No | 5062 | 69.7 | <0.001 | | | 0.90 (0.86 to 0.94) |
| Yes | 2196 | 30.3 | 0.003 | | | 0.89 (0.83 to 0.96) |
| social activities | | | | 0.08 | | |
| No | 3003 | 41.4 | <0.001 | | | 0.86 (0.82 to 0.91) |
| Yes | 4255 | 58.6 | 0.001 | | | 0.92 (0.88 to 0.97) |
| Pain | | | | 0.862 | | |
| No | 6062 | 83.5 | <0.001 | | | 0.89 (0.86 to 0.93) |
| Yes | 1196 | 16.5 | 0.014 | | | 0.90 (0.82 to 0.98) |
| Overall | 7258 | 100 | <0.001 | | | 0.89 (0.86 to 0.93) |

**Fig 3. Subgroup analysis of the association between BRI and the risk of depression.**

structure, reduced emotional processing capabilities, and behavioral abnormalities [33]. For example, obesity can affect the synthesis, release, and transport of neurotransmitters like serotonin (5-HT) and dopamine, thereby influencing emotional regulation and reward mechanisms [34]. Considering age-related physiological changes such as decreased antioxidant

capacity, HPA axis dysfunction, and reduced gut microbiota diversity, middle-aged and older adults are more prone to these mechanisms, thereby increasing their risk of developing depression. Therefore, the association of obesity with depression was particularly pronounced in the middle-aged and elderly population, BRI, an obesity indicator that integrates height and WC, BRI can more accurately reflect the obesity status of this demographic and its relationship with the risk of depression.

Obesity and depression may affect each other. On the one hand, obesity can raise depression risk through inflammation and HPA axis dysfunction [27]. On the other hand, individuals with depression often experience appetite disorders (such as emotional eating), reduced physical activity, and metabolic side effects of antidepressants (such as weight gain), which could further exacerbate obesity. This bidirectional relationship suggests that the association between BRI and depressive symptoms may partially reflect the mutual reinforcement between the two conditions. Therefore, future research should employ methods such as cross-lagged panel models or mediation analysis to elucidate the temporal sequences and potential mediating mechanisms between obesity and depression.

This study presents several advantages. Firstly, it leverages high-quality longitudinal data from the CHARLS, employing PPS random sampling method to ensure broad representativeness of the sample, thereby enhancing the external validity of the findings. Secondly, this study utilized RCS and threshold effect analysis to systematically evaluate the association between BRI and depression. In clinical settings, BRI can be employed as an extra tool to check for depression risk. Since it's easy to calculate, BRI works well in primary care places and health check-ups. Clinical practitioners can integrate BRI into routine health assessments for middle-aged and older adults, complementing these assessments with mental health scales such as CESD-10 for comprehensive screening. According to the established BRI threshold (4.29), the middle-aged and older adults can be categorized into two groups: low risk of depression (BRI < 4.29) and high risk of depression (BRI ≥ 4.29). For the low-risk group, routine health education and regular follow-up are sufficient. In contrast, for high-risk individuals, it is advisable to intensify monitoring of depressive symptoms and implement early intervention strategies, including psychological counseling and lifestyle recommendations. Furthermore, BRI can be used in conjunction with traditional indicators such as BMI and WC. In comparison to BMI, BRI more accurately reflects the distribution of abdominal fat, aiding in the identification of depression-prone individuals who are normal weight but have abdominal obesity.

This study also has several limitations. Firstly, while a longitudinal design was employed to investigate the predictive effect of baseline BRI on the risk of depression, relevant covariates have been adjusted in the analysis. However, there may still be some residual confounding factors that have not been fully controlled, potentially impacting the research results. Secondly, although participants with pre-existing depression were excluded at baseline, subclinical depression or mild depressive symptoms may have influenced BRI levels during follow-up, thereby affecting the inference of the directionality of the association between the two. Finally, this study only focused on baseline BRI levels and did not include longitudinal dynamic changes in BRI data. This limitation may restrict our understanding of the connection between BRI evolution and depression risk. Future research should establish a more long-term tracking mechanism to systematically explore the trajectory of BRI over time and its dynamic correlation patterns with the development of depressive symptoms.

## 5. Conclusion

Our findings reveal the notable association between higher BRI levels and an increased risk of developing depression in the middle-aged and elderly Chinese population. This suggests that BRI could serve as a simple and readily accessible indicator for predicting depression risk.

Future research should aim to elucidate the underlying biological mechanisms connecting BRI and depression, as well as evaluate the potential role of regulating BRI levels in the prevention and treatment of depression.

## Supporting information

**S1 data.**
(XML)

**S2 File.  Explanation of reported items of the STROBE checklist.** The STROBE Statement can be obtained and assessed through the STROBE website [10] or the article by von Elm et al. [35]. For further details on its evaluation framework and criteria, refer to the comprehensive interpretation by Vandenbroucke et al [36]. The quality of reporting for each STROBE item is rated as "adequately reported," "inadequately reported," or "not applicable." Five items (6a, 6b, 12d, 14c, and 15) are related to specific study designs. If an item is not applicable, it is rated as "not applicable" and is not included in the total number of items to be evaluated.
(DOCX)

## Author contributions

**Conceptualization:** Wenfei Yang, LiPing Chen, Liling Tong.

**Formal analysis:** Wenfei Yang, LiPing Chen, Liling Tong.

**Funding acquisition:** Liling Tong, Wenchang He, Hua Lin.

**Supervision:** LiPing Chen, Liling Tong.

**Validation:** Hua Lin.

**Writing – original draft:** Wenfei Yang, Liling Tong, Wenchang He.

**Writing – review & editing:** Wenfei Yang, Liling Tong, Wenchang He, Hua Lin.

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
