## [Decision Letter · Decision Letter 0]

28 Jan 2025

PONE-D-24-55220Association between Body Roundness Index and Depression Among Middle-aged and Older Adults in Chinese Communities: An Empirical Analysis Based on CHARLS DataPLOS ONE

Dear Dr. Lin,

Thank you for submitting your manuscript to PLOS ONE. After careful consideration, we feel that it has merit but does not fully meet PLOS ONE’s publication criteria as it currently stands. Therefore, we invite you to submit a revised version of the manuscript that addresses the points raised during the review process. Please submit your revised manuscript by Mar 14 2025 11:59PM. If you will need more time than this to complete your revisions, please reply to this message or contact the journal office at plosone@plos.org. Please include the following items when submitting your revised manuscript:

We look forward to receiving your revised manuscript.

Kind regards,

Fahad Farhan Almutairi, PhD

Academic Editor

PLOS ONE

2. Thank you for stating the following financial disclosure:  [This study was supported by the Hainan Province Department of Science and Technology Development Project (Project Number: 822RC864).]. 

3. Please include a copy of Table 1, 2, 3 which you refer to in your text on page 8.

Additional Editor Comments (if provided):

Reviewers' comments:

Reviewer's Responses to Questions

**Comments to the Author**

1. Is the manuscript technically sound, and do the data support the conclusions?

Reviewer #1: Yes

Reviewer #2: Yes

2. Has the statistical analysis been performed appropriately and rigorously? 

Reviewer #1: Yes

Reviewer #2: Yes

3. Have the authors made all data underlying the findings in their manuscript fully available?

Reviewer #1: Yes

Reviewer #2: Yes

4. Is the manuscript presented in an intelligible fashion and written in standard English?

Reviewer #1: Yes

Reviewer #2: No

5. Review Comments to the Author

Reviewer #1: Thank you for the chance to review this paper.

The study investigates a very important topic. The methods are good, and the results are described and discussed well.

The only comment that I have is regarding the tool used. The authors didn't specify what language the survey was presented in and, if it was translated, what methods were used to test the validity of the tool after the translation.

There are a few typos that should be checked.

Reviewer #2: Review Comments

Congratulations to the authors and the team on completing this manuscript. This study addresses a significant public health issue by exploring the relationship between Body Roundness Index (BRI) and depression risk among older Chinese adults. The use of robust statistical methods and a large dataset enhances the credibility of the findings. However, there are areas where further refinement would improve the clarity, rigor, and impact of the manuscript. Below are detailed comments and suggestions for each section.

General Comments

Abbreviations: While many abbreviations (e.g., BMI, BRI, CHARLS) may be familiar to readers, it is recommended to present the full term followed by the abbreviation upon first use. Please ensure consistency throughout the manuscript.

Language and Grammar: Several minor spelling and grammatical errors need attention, such as "behavio" in line 41, an extra ")" in line 63, and "Educational leve" in the tables. Line 136 ("were included?") and other grammatical issues also require correction. A thorough proofreading and professional English editing are strongly recommended. Please include an English editing certificate upon resubmission.

Introduction

The introduction is well-organized and provides a comprehensive overview of the topic, but it could be more focused and streamlined to highlight the core research question better.

Suggestions for Improvement

1. Focus on the Core Theme: The connection between BRI and depression, the central research question, is introduced too late in the introduction. Consider reorganizing the content to bring this focus forward.

2. Streamline Discussion on BMI: The section discussing BMI’s limitations is overly detailed and could be condensed to maintain narrative flow.

3. Strengthen Transitions: Improve transitions between sections to make the progression from depression to BMI, and then to BRI, more seamless.

4. Highlight Research Significance: Clearly articulate why this study focuses on older Chinese adults and how it addresses gaps in the existing literature.

Proposed Structure for Rewriting

1. First Paragraph: Briefly introduce depression and its association with obesity, emphasizing the public health relevance.

2. Second Paragraph: Discuss BMI as a commonly used but limited measure of obesity, highlighting the need for alternative indicators like BRI.

3. Third Paragraph: Introduce BRI, its advantages over BMI, and its potential relationship with depression. Conclude by stating the research objectives and hypotheses.

Materials and Methods

The methods section is well-structured and provides detailed information about the dataset and statistical approaches. However, some clarifications and additional details would improve transparency.

Suggestions for Improvement

1. Data Source and Sample Selection: Clarify why Wave 3 and Wave 5 of the CHARLS dataset were chosen and address any potential selection bias.

2. Line 102: Include the CHARLS website URL as a reference.

3. Depression Measurement: Specify whether the CESD-10 scale used was the English or Chinese version. If the Chinese version was used, cite validation studies ensuring its reliability and validity. If none exist, mention this as a limitation and cite the original CESD-10 study.

4. Anthropometric Measurements: Indicate whether height, weight, and waist circumference were self-reported or measured by professionals. If measured, provide details about the equipment or personnel involved.

5. Clarify Gender and Sex: Use "gender" and "sex" appropriately throughout the manuscript, as these terms are not interchangeable in academic contexts.

6. Wave Consistency: Line 147: Confirm whether the 2015 baseline data correspond to Wave 1 or Wave 3, as the CHARLS website indicates Wave 1 refers to 2011. Clarify this discrepancy.

7. Explain the rationale for using Wave 3 BRI to predict Wave 5 depression risk without considering Wave 5 BRI. Discuss whether changes in BRI between waves might influence the results.

8. Rewrite the sentence: ”Statistical analyses were conducted using R software (version X.X.X).”

Results

The results are presented clearly, with appropriate statistical models. However, there are areas where additional context and interpretation are needed to enhance understanding.

Suggestions for Improvement

1. Annotations in Tables and Figures: Ensure tables and figures can be understood independently. Provide full terms for abbreviations in table footnotes and include explanations of clinical or practical significance for the results (e.g., OR values and 95% CIs in Table 2). Clarify unclear elements, such as "P P for overall" in Figure 2.

2. Nonlinear Relationships: In Figure 2, annotate the key threshold (e.g., 4.29) and explain its significance for interpreting BRI’s impact on depression risk.

3. Model Fit: Include model fit indices (e.g., AIC, BIC) to justify and confirm the chosen models are optimal.

4. Dynamic Changes in BRI: Address whether changes in BRI between Wave 3 and Wave 5 influence depression risk. If fluctuations exist, discuss their potential implications.

Discussion

The discussion provides valuable insights into potential mechanisms linking BRI to depression but could benefit from additional depth and broader contextualization.

Suggestions for Improvement

1. Causal Inference: It would be beneficial to contextualize the limitations of causal inference further. Highlight the limitations of cross-sectional data and recommend more rigorous longitudinal or interventional studies to validate the findings. Consider the bidirectional relationship between BRI and depression, where depression might also contribute to obesity. Discuss how this bi-directionality could influence interpretation.

2. International Context: Compare findings with similar studies in other populations or regions to emphasize this study's novelty and broader relevance.

3. Mechanistic Hypotheses: Expand on the proposed mechanisms (e.g., inflammation, HPA axis dysregulation, gut microbiota) with more recent literature. Explain why these mechanisms are particularly relevant to middle-aged and older Chinese adults.

4. Practical Applications: Discuss how BRI could be integrated into clinical practice or public health policies, including the feasibility of using the 4.29 threshold in depression screening programs.

Final Remarks

Congratulations to the authors and the team for studying such an important topic. The use of an obesity index (BRI) in the context of depression risk is relevant. Addressing the above suggestions will help refine the manuscript, enhance its clarity and rigor, and increase its contribution to the field in China. I encourage the authors to continue this valuable research and look forward to seeing its further development.

6. PLOS authors have the option to publish the peer review history of their article (what does this mean?). If published, this will include your full peer review and any attached files.

Reviewer #1: No

Reviewer #2: No

---

## [Author Response · Author response to Decision Letter 1]

11 Feb 2025

Dear Editor and reviewers,

We sincerely appreciate the time you dedicated to providing constructive feedback on our manuscript. Your insightful comments have significantly enhanced its quality. In response to your suggestions, we have made the necessary changes, which are highlighted in red in the manuscript and supplementary materials. Below, we present a point-by-point response to the comments from both the editor and the reviewers. Thank you once again for your valuable input.

GENERAL COMMENT

1. Abbreviations: While many abbreviations (e.g.BMI, BRI, CHARLS) may be familiar to readers, it is recommended to present the full term followed by the abbreviation upon first use. Please ensure consistency throughout the manuscript.

Author Response: We greatly appreciate your valuable feedback on our manuscript. Concerning the abbreviation standardization issue you raised, we have comprehensively revised the terms such as BMI, BRI, and CHARLS according to the "full name - abbreviation" format principle. This ensures that all abbreviations are accompanied by their complete spellings upon their first appearance and carefully checked the consistency throughout the text to comply with standardized expression.

2. Language and Grammar: Several minor spelling and grammatical errors need attention, such as "behavio" in line 41, an extra ")" in line 63, and "Educational leve" in the tables. Line 136 ("were included?") and other grammatical issues also require correction. A thorough proofreading and professional English editing are strongly recommended. Please include an English editing certificate upon resubmission.

Author Response: We greatly appreciate your valuable feedback on our manuscript.

Regarding language and grammar: We have carefully proofread the entire manuscript based on your feedback and corrected all the specified spelling and grammatical errors, including the word “behavio” in line 41, the extra parentheses in line 63, and the “educational term” in the tables, as well as the grammatical issues mentioned in line 136 and other similar instances. To further enhance the language quality, we have engaged a professional English editing service and have attached the editing certificate. All revised content is highlighted in red.

REVIEWER 1 EVALUATION

1. The only comment that I have is regarding the tool used. The authors didn't specify what language the survey was presented in and, if it was translated, what methods were used to test the validity of the tool after the translation. There are a few typos that should be checked.

Author Response: We sincerely appreciate the valuable comments on our manuscript, particularly your attention to detail and the time spent on the review. Regarding the assessment tool for depression that you mentioned, the CES-D-10 is a brief self - rating scale for depressive symptoms, primarily used to screen for depressive symptoms in various populations, including the elderly. The Chinese version adopted in the CHARLS database has been proven to have good reliability and validity. We have also re - examined the reliability and validity data of the original CESD-10S scale and related literature has been cited in this paper. (please refer to line 120-131 on page 6 of the revised manuscript).

Regarding spelling errors, we have carefully reviewed and corrected all the errors identified throughout the manuscript and conducted additional proofreading to minimize the possibility of any residual mistakes. Moreover, we have invited a native English speaker to polish our article.

REVIEWER ２ EVALUATION

1. Suggestions for Improvement

1.1 Focus on the Core Theme: The connection between BRI and depression, the central research question, is introduced too late in the introduction. Consider reorganizing the content to bring this focus forward.

Author Response: Thank you for your suggestions. According to your suggestion，We have revised the background introduction, introducing the correlation between BRI and depression earlier to highlight the core research question more effectively and focus on the elaboration of their relationship.

1.2 Streamline Discussion on BMI: The section discussing BMI’s limitations is overly detailed and could be condensed to maintain narrative flow.

Author Response: Thank you for your suggestions. We streamlined the discussion on the limitations of BMI, retained core points and removed redundant information. Thanks again for helping to enhance the clarity and fluency of the manuscript! The revisions have been highlighted in the document, and I look forward to your further feedback.

1.3 Strengthen Transitions: Improve transitions between sections to make the progression from depression to BMI, and then to BRI, more seamless.

Author Response: Thank you for your valuable comments on the logical coherence of our paper. We fully recognize the significant impact of paragraph transitions on readability. During the revision process, we systematically reviewed the logical progression from “depression to BMI to BRI,” and added transitional sentences and explanatory phrases to enhance the smoothness of our arguments.

1.4 Highlight Research Significance: Clearly articulate why this study focuses on older Chinese adults and how it addresses gaps in the existing literature.

Author Response: Thank you for your crucial suggestions on the positioning of the study's innovation. We fully agree with the academic value of clarifying the uniqueness of the study population. In the last paragraph of the Introduction, we have emphasized the reasons for focusing on middle-aged and older adults in China. The population of older adults in China is expanding at a rapid pace, and this demographic is particularly vulnerable to depression, which can exert a profound impact on their physical health, cognitive function, and overall quality of life. Individuals aged 60 and above in China face a higher risk of depression, with 33.1% exhibiting depressive symptoms and 5.3% suffering from major depressive disorder. By examining the association between BRI and depressive symptoms, we aim to identify potential modifiable risk factors that could be targeted in prevention and treatment strategies.

However, most existing studies are based on Western populations, and their conclusions are difficult to directly apply to the Chinese elderly population, which is influenced by unique lifestyles, sociocultural contexts (such as family structure, social support systems, and cultural values). Therefore, our study fills this gap by conducting research on this specific population.

2. Proposed Structure for Rewriting

2.1 First Paragraph: Briefly introduce depression and its association with obesity, emphasizing the public health relevance.

Author Response: Thank you for your suggestions. We have revised the content according to your suggestions, strengthened the introduction of depression and its relationship with obesity, and emphasized its relevance to public health. (Please refer to lines 47-62 on page 3).

For your convenience, the revised content is also provided here:

Depression and obesity are global health challenges that impact public health. The main features of depression include persistent mood disturbances, often accompanied by cognitive impairment, and deterioration of physical and social functioning. Severe cases may lead to suicide [1]. Depression affects patients' quality of life, social functioning, and creates a large economic burden for families and society [2]. The World Health Organization reports that about 350 million people global suffer from depression, with an increasing incidence rate yearly [3]. In China, about 33.1% of adults aged 60 and above show depression symptoms, with a prevalence of major depressive disorder reaching 5.3% [4]. Elderly people are more likely to suffer from depression due to declining physiological functions, social role changes, and increased chronic disease rates [5]. The causes of depression involve genetics, neurobiology, and social influences [6, 7]. Additionally, depression exhibits high heterogeneity and uncertainty in symptoms, progression, prognosis, and treatment response, making treatment and management challenging [8]. Therefore, strengthening early identification and prevention of depression is crucial for effectively curbing the disease progression.

2.2 Second Paragraph Discuss BMI as a commonly used but limited measure of obesity, highlighting the need for alternative indicators like BRI.

Author Response: Thank you for your suggestions. We have revised the content according to your suggestions, streamlined the introduction of obesity, and enhanced the discussion on the limitations of BMI, highlighting the necessity of exploring alternative indicators such as BRI to more comprehensively assess obesity and its related health risks. (Please refer to lines 63-67 on page 3 and lines 68-76 on page4 ).

For your convenience, the revised content is also provided here:

Evidence shows that obesity is an important risk factor for depression. Obese individuals have a 1.29 to 1.69 times higher risk of depression than those with normal weight [9]. Traditionally, Body Mass Index (BMI) and Waist Circumference (WC) are commonly used to assess obesity, but both WC and BMI have their limitations in evaluating obesity.BMI cannot distinguish fat distribution from muscle mass, and WC as an indicator of obesity assessment also has its inherent limitations [10]. In addition, There is a complex nonlinear association between obesity and mortality, called the "obesity paradox" [11], which challenges the application of traditional obesity indicators in health risk assessment. Studies on the association between obesity and depression show complexity, with some suggesting obesity increases depression risk [12] and others suggesting a protective effect [13]. This inconsistency highlights the need for a more detailed evaluation of traditional indicators like BMI and calls for the exploration of more accurate methods for measuring obesity phenotypes.

2.3 Third Paragraph Introduce BRI, its advantages over BMI, and its potential relationship with depression. Conclude by stating the research objectives and hypotheses.

Author Response: Thank you for your suggestions. We have revised the relevant sections according to your suggestions, providing a clearer introduction to the BRI and elaborating on its advantages over the BMI and its potential link to depression. We have highlighted the strengths of BRI in capturing body fat distribution, especially central obesity. Additionally, we have summarized the preliminary studies supporting the potential association between BRI and depression and clearly stated the objective of our study: to investigate whether BRI can more effectively predict depression risk than BMI. (Please refer to lines 77-90 on page 4).

For your convenience, the revised content is also provided here:

The body roundness index (BRI) is a new measure of obesity that combines WC and height, providing a more accurate reflection of abdominal fat accumulation and individual body characteristics [14]. Compared to BMI, BRI can more accurately assess the severity of obesity [15]. Studies have demonstrated that BRI can effectively forecast the risk of chronic conditions [16,17]. A study based on the National Health and Nutrition Examination Survey (NHANES) revealed a positive correlation between BRI and depression in adults [18], but the applicability of this conclusion in middle-aged and older adults still requires further validation. Given the accelerated aging of China's population and the rising prevalence of depression among middle-aged and older adults, it is important to explore the connection between BRI and depressive symptoms in this demographic. Therefore, this research seeks to explore the relationship between BRI and depression using data from the China Health and Retirement Longitudinal Study (CHARLS), providing scientific evidence for the prevention and intervention of obesity and depression.

3. Materials and Methods

3.1 The methods section is well-structured and provides detailed information about the dataset and statistical approaches. However, some clarifications and additional details would improve transparency. Data Source and Sample Selection: Clarify why Wave 3 and Wave 5 of the CHARLS dataset were chosen and address any potential selection bias.

Author Response: Thank you for your suggestions. When selecting the Wave 3 (2015) and the wave 5 (2020) of the CHARLS dataset to explore the association between the Body Roundness Index (BRI) and depressive symptoms among middle-aged and older Chinese adults, our consideration was primarily based on the long-term development characteristics of depressive symptoms. Depressive symptoms are often the result of a long-term process, shaped by the interaction of multiple factors over an extended period. Therefore, to thoroughly investigate this association, a dataset with a sufficient time span is necessary. The 5-year interval provides a valuable temporal window that allows us to capture the long-term and subtle relationships between depressive symptoms and BRI.

For the convenience of reviewers and readers, we have revised the relevant paragraph to read: “This study utilizes the CHARLS dataset, with the 2015 baseline data and the 2020 follow-up data for longitudinal analysis.”

3.2 Line 102: Include the CHARLS website URL as a reference. https://charls.pku.edu.cn/

Author Response: Thank you for your suggestions. We have added the URL of the CHARLS website in the references.: https://charls.pku.edu.cn/(Please refer to lines 104 on page 5).

3.3 Depression Measurement: Specify whether the CESD-10 scale used was the English or Chinese version. If the Chinese version was used, cite validation studies ensuring its reliability and validity. If none exist, mention this as a limitation and cite the original CESD-10 study.

Author Response: We sincerely appreciate the valuable comments on our manuscript. Regarding the assessment tool for depression that you mentioned, the CES-D-10 is a brief self - rating scale for depressive symptoms, primarily used to screen for depressive symptoms in various populations, including the elderly. The Chinese version adopted in the CHARLS database has been proven to have good reliability and validity. We have also re - examined the reliability and validity data of the original CESD-10S scale and its translated versions and cited relevant literature to substantiate this (please refer to lines 120 - 131 on page 6 ).

3.4 Anthropometric Measurements: Indicate whether height, weight, and waist circumference were self-reported or measured by professionals. If measured, provide details about the equipment or personnel involved.

Author Response: Thank you for your suggestions. Regarding your query about the height and waist circumference measurement methods used in the CHARLS database, we have verified the procedures and provide the following clarification:

Height Measurement:

For height measurement, a wall-mounted stadiometer was employed. Participants stood barefoot with heels together, toes angled approximately 60 degrees, and back fully contacting the vertical board. The head was positioned in the Frankfort horizontal plane (aligned with the ear tragus and lower orbital rim). Trained technicians gently lowered the horizontal headplate until firm contact with the crown was achieved, with measurements recorded to the nearest 0.1 cm.

Waist Circumference Measurement:

Waist circumference was measured using a non-elastic tape. Participants maintained a relaxed standing posture with feet together and arms hanging naturally. The tape was positioned horizontally at the midpoint between the lowest rib margin and iliac crest (corresponding to the natural waistline), ensuring snug contact without compressing subcutaneous tissue. Measurements were taken at the end of normal expiration and recorded to the nearest 0.1 cm.

We hope this clarification adequately addresses your concerns and demonstrates our commitment to methodological transparency. Should additional details be required, we will provide supplementary documentation from the CHARLS operations manua (Please refer to lines 135 on page 6).

3.5 Clarify

---

## [Editor Report · Decision Letter 1]

14 Feb 2025

Association between Body Roundness Index and Depression Among Middle-aged and Older Adults in Chinese Communities: An Empirical Analysis Based on CHARLS Data

PONE-D-24-55220R1

Dear Dr. Lin,

We’re pleased to inform you that your manuscript has been judged scientifically suitable for publication and will be formally accepted for publication once it meets all outstanding technical requirements.

Kind regards,

Fahad Farhan Almutairi, PhD

Academic Editor

PLOS ONE

---

## [Editor Report · Acceptance letter]

PONE-D-24-55220R1

PLOS ONE

Dear Dr. Lin,

I'm pleased to inform you that your manuscript has been deemed suitable for publication in PLOS ONE. Congratulations! Your manuscript is now being handed over to our production team.

Kind regards,

on behalf of

Dr. Fahad Farhan Almutairi

Academic Editor

PLOS ONE